# Influence of gravity waves on the climatology of high-altitude Martian carbon dioxide ice clouds

Erdal Yiğit[1], Alexander S. Medvedev[2], and Paul Hartogh[2]

[1]Department of Physics and Astronomy, George Mason University, Fairfax, VA, USA
[2]Max Planck Institute for Solar System Research, Göttingen, Germany

**Correspondence:** Erdal Yiğit (eyigit@gmu.edu)

**Abstract.** Carbon dioxide ($CO_2$) ice clouds have been routinely observed in the middle atmosphere of Mars. However, there are still uncertainties concerning physical mechanisms that control their altitude, geographical, and seasonal distributions. Using the Max Planck Institute Martian General Circulation Model (MPI-MGCM), incorporating a state-of-the-art whole atmosphere subgrid-scale gravity wave parameterization (*Yiğit et al.*, 2008), we demonstrate that internal gravity waves generated by lower atmospheric weather processes have wide reaching impact on the Martian climate. Globally, GWs cool the upper atmosphere of Mars by $\sim$10% and facilitate high-altitude $CO_2$ ice cloud formation. $CO_2$ ice cloud seasonal variations in the mesosphere and the mesopause region appreciably coincide with the spatio-temporal variations of GW effects, providing insight into the observed distribution of clouds. Our results suggest that GW propagation and dissipation constitute a necessary physical mechanism for $CO_2$ ice cloud formation in the Martian upper atmosphere during all seasons.

## 1 Introduction

Mars is the second most studied terrestrial planet due to its similarity and also differences to Earth. For example, Mars is half the size of Earth, has two very exotic dwarf satellites, Phobos and Deimos; has a similar to Earth orbital tilt, and it takes Mars nearly 1.9 Earth years to go around Sun, with a much larger eccentricity than Earth (Appendix A). Thus, Mars has seasons similar to those on Earth. Studying Mars can serve, beside the aspects of habitability, as a natural fluid dynamical laboratory, where geophysicists can test the understanding and applicability of basic fluid dynamical principles.

Carbon dioxide clouds have been routinely observed in the Martian atmosphere at various altitudes between $\sim$50 and $\sim$100 km (*Clancy and Sandor*, 1998; *Clancy et al.*, 2007; *Colaprete et al.*, 2008; *Määttänen et al.*, 2010; *McConnochie et al.*, 2010; *Vincendon et al.*, 2011; *González-Galindo et al.*, 2011; *Määttänen et al.*, 2013; *Sefton-Nash et al.*, 2013; *Stevens et al.*, 2017; *Aoki et al.*, 2018). It was hypothesized that these so-called high-altitude clouds are formed in the regions where temperature drops below the $CO_2$ condensation threshold, which were first detected in the mesosphere during Mars Pathfinder entry and descent (*Schofield et al.*, 1997). These high-altitude clouds are to some extent analogous to the noctilucent clouds (NLCs) observed in Earth's mesosphere (*Witt*, 1962), which are indeed high-altitude clouds. Previous numerical simulations and observations showed that gravity wave-induced dynamical effects, such as wind fluctuations lead to the structures observed in NLCs (*Jensen and Thomas*, 1994; *Rapp et al.*, 2002).

Because the   Martian mesosphere is in general warmer than the condensation threshold, *Clancy and Sandor* (1998) suggested that clouds can form in pockets of cold air created occasionally by a superposition of fluctuations associated with solar tides and gravity waves (GWs). Certainly, cold temperatures are not the only physical mechanism required for $CO_2$ cloud formation. The microphysics calculations demonstrated the dependence of nucleation processes on the existence and sizes of condensation nuclei (*Määttänen et al.*, 2010), and that temperature excursions of several to tens of Kelvins below the condensation threshold are required. Simulations with the Laboratoire de Météorologie Dynamique Martian general circulation model (LMD-MGCM) demonstrated that the spatial and temporal distributions of the predicted cold temperatures generally correlated with the observations of high-altitude $CO_2$ clouds, but could not reproduce all of their features (*González-Galindo et al.*, 2011). This study revealed the role of thermal tides in cloud formations and the authors suggested that the discrepancies could be caused by the neglect of GWs, which were neither resolved by the MGCM, nor accounted for in a parameterized form (with the exception of harmonics with zero   horizontal phase velocities with respect to the surface generated by the flow over topography). The role of GWs was further addressed in the work by *Spiga et al.* (2012), who used a mesoscale (GW-resolving) limited area model to demonstrate for the first time with direct simulations that orographically generated waves can propagate to the mesosphere and facilitate a creation of cold air patches at supersaturated temperatures. They compared the distribution of a linear wave saturation index with observed clouds to find that the latter reasonably well coincided with regions where GWs had favorable propagation conditions.

The next step in the attempt to explain the observations of high-altitude $CO_2$ clouds on the globe has been performed with the Max Planck Institute (MPI) MGCM coupled with a whole atmosphere GW parameterization (*Yiğit et al.*, 2015a). It was shown that this technique can reproduce the occurrences of supersaturated temperatures in low latitudes during a vernal equinox, in a good agreement with observations of mesospheric $CO_2$ clouds, which however, distinctively vary with seasons (e.g., *González-Galindo et al.*, 2011; *Sefton-Nash et al.*, 2013). In particular, the observational study of *Sefton-Nash et al.* (2013) using data from NASA's Mars Climate Sounder (MCS) onboard Mars Reconnaissance Orbiter (MRO), which demonstrated that the high-altitude clouds are continuously present in the Martian atmosphere with distinct seasonal and latitudinal behavior. From a theoretical standpoint, it is thus instructive to study the seasonal behavior of $CO_2$ clouds in order to gain insight into the underlying processes. In this paper, we extend the approach with parameterized GWs to further assess the role of small-scale GWs in shaping spatial and seasonal variations of pockets of cold air, which are pre-requisites for $CO_2$ cloud formation (*Listowski et al.*, 2014).

Propagation of GWs into the thermosphere has been studied extensively for Earth using idealized wave models (e.g., *Hickey and Cole*, 1988; *Walterscheid et al.*, 2013) and general circulation models (GCMs) (e.g., *Yiğit et al.*, 2009; *Miyoshi et al.*, 2014). They explored the fundamental processes that control propagation and dissipation of a broad spectrum of internal waves (*Yiğit and Medvedev*, 2015). On Mars, numerical wave models demonstrated that GWs can propagate into the upper atmosphere and produce similar significant dynamical and thermal forcing there (*Parish et al.*, 2009). In particular, implementation of the whole atmosphere GW scheme of *Yiğit et al.* (2008) into the Max Planck Institute Martian General Circulation Model (MPI-MGCM) revealed substantial dynamical effects (i.e., acceleration/deceleration) in the Martian upper mesosphere and lower

thermosphere around 90–130 km, in the region of interest of this study (*Medvedev et al.*, 2011a). Recently, upper atmospheric signatures of small-scale GW waves have routinely been observed (*Yiğit et al.*, 2015b; *England et al.*, 2017).

The structure of our paper is as follows: Next section describes the methods utilized in this research, describing the MPI-MGCM, the whole atmosphere GW parameterization and the link between clouds and waves; section 3 presents an analysis of the global annual mean fields; sections 4 and 5 analyze the seasonal variations of the mean fields, gravity wave activity, and $CO_2$ cloud formation. Section 6 discusses simulation results in the context of previous research and observations. Summary and conclusions are given in section 7.

## 2    Methodology

We next describe the MGCM, outline the implemented whole atmosphere GW parameterization, how it is linked to $CO_2$ cloud formation in the model, and the setup of numerical experiments.

### 2.1    Martian General Circulation Model (MGCM)

The Max Planck Institute Martian General Circulation Model (MPI-MGCM) calculates a three-dimensional time-dependent evolution of the horizontal and vertical winds, temperature and density of the neutral atmosphere by solving the momentum, energy and continuity equations on a globe. The present state of the model is the result of incremental historical development. It contains the physical parameterizations of the earlier versions (*Hartogh et al.*, 2005, 2007; *Medvedev and Hartogh*, 2007) and the spectral dynamical solver introduced in the work of *Medvedev et al.* (2011b). Of particular relevance to the subject of this paper are the parameterizations of $CO_2$ condensation/sublimation and the radiative heating/cooling scheme due to IR transfer by $CO_2$ molecules under the breakdown of the local thermodynamic equilibrium (non-LTE). The former accounts for phase transitions, sedimentation of ice particles, surface ice accumulation and seasonal polar ice caps, thermal and mass effects. In the latter, the atomic oxygen profile of *Nair et al.* (1994) and the $CO_2$-O quenching rate coefficient $k_{\nu T} = 3.0 \cdot 10^{-12} \mathrm{cm}^3 \, \mathrm{s}^{-1}$ were used, as described in the paper of *Medvedev et al.* (2015).

The simulations have been performed with the T21 horizontal spectral truncation, which corresponds to $64 \times 32$ grid point resolution in longitude and latitude, corresponding to approximately $5.5° \times 5.5°$ resolution, respectively. The current version of the model uses 67 hybrid vertical coordinates (terrain-following in the lower atmosphere gradually and changing to pressure-based in the upper atmosphere). Its domain extends into the thermosphere to $3.6 \times 10^6$ Pa (150–160 km, depending on solar activity, temperature, etc).

### 2.2    Whole Atmosphere Gravity Wave Parameterization

GCMs typically have resolutions insufficient for reproducing small-scale GWs. Therefore, the influence of subgrid-scale GWs on the larger-scale atmospheric circulation has to be parameterized. The parameterizations then estimate the effects of unresolved GWs on the resolved, large-scale flow using first principles. The vast majority of GW schemes have been designed for terrestrial middle atmosphere GCMs (*Fritts and Alexander*, 2003, see Sect. 7) and, thus, are not well suited for dissipa-

tive media such as Earth's thermosphere and Mars' middle and upper atmosphere. We employ a GW parameterization that is specifically developed to overcome this limitation. It was described in detail in the work of *Yiğit et al.* (2008), and the general principles of the extension of GW parameterizations into whole atmosphere schemes have been discussed later in the work by *Yiğit and Medvedev* (2013). This scheme has extensively been tested for the terrestrial environment, e.g., see the works by *Yiğit and Medvedev* (2016) and *Yiğit and Medvedev* (2017) for the recent application with the Coupled Middle Atmosphere Thermosphere-2 (CMAT2) model. The parameterization was also used within the MPI-MGCM (see e.g., *Medvedev et al.*, 2015, 2016, for recent applications) and has recently been tested in a Venusian GCM (*Brecht et al.*, 2018).

Physically-based parameterizations usually rely on certain simplifications. In the GW scheme applied here, information about wave phases is neglected, while covariances, including the squared amplitude, are still evaluated. In particular, the scheme calculates the vertical evolution of the vertical flux of GW horizontal momentum, $\overline{u'w'}(z) = (\overline{u'w'}, \overline{v'w'})$, taking account for the effect of dissipation on a broad spectrum of GW harmonics. In the middle and upper atmosphere of Mars, wave damping occurs due primarily to nonlinear wave-wave interactions (breaking and/or saturation) and molecular diffusion and thermal conduction, which are accounted for through the transmissivity $\tau_i$ (*Yiğit et al.*, 2009):

$$\overline{u'w'}_i(z) = \overline{u'w'}_i(z_0) \frac{\rho(z_0)}{\rho(z)} \tau_i(z). \tag{1}$$

Here overbars denote an appropriate averaging, the subscript $i$ indicates a given GW harmonic, $\overline{u'w'}_i(z_0)$ are the fluxes at a certain source level $z_0$, and $\rho$ is the mass density. This formulation requires also a prescription of the characteristic horizontal scale $\lambda_h$ of GWs for calculating $\tau_i$. For the reasons ~~multiply~~ described in our papers (e.g., see the last paragraph of Section 4 of *Medvedev et al.*, 2011a), $\lambda_h = 300$ km was adopted in the simulations. Unlike in many conventional GW schemes, no additional intermittency factors, which are often regarded as tuning factors, are used in our scheme, because the latter is included in averaging. The parameterization is called "spectral", because it considers propagation of a broad spectrum of waves with different horizontal phase velocities $c_i$ (or vertical wavelengths). The initial momentum fluxes of the phase speeds have a Gaussian distribution (*Medvedev et al.*, 2011a, Figure 2). Note that orographically-generated GWs are represented by a single harmonic $c = 0$. The scheme takes account of interactions between GW harmonics, rather than considering them as a mere superposition of propagating waves. Therefore, it is sometimes called "nonlinear". Finally, the parameterization is characterized as a "whole atmosphere" one to signify its physical applicability to all atmospheric layers.

The available observational constraints on GW sources in the lower atmosphere of Mars have been discussed in the work of *Medvedev et al.* (2011b). First, we assume a horizontally-uniform total momentum fluxes in the troposphere with the maximum magnitude of 0.0025 m$^2$ s$^{-2}$. Recent simulations with a high-resolution MGCM (*Kuroda et al.*, 2015, 2016) demonstrated that the sources of small-scale waves strongly vary horizontally and with seasons and can significantly exceed this value. Thus, the current setup allows for capturing only mean GW effects and not full details. Second, there is a lack of detailed knowledge of GW spectra in the Martian atmosphere. Meanwhile, there are indications of "universality" of these spectra (*Ando et al.*, 2012). Thus, we assume the similar spectral shape of GWs in the troposphere as on Earth. Third, we consider that the mean wind at the source level modulates the direction of propagation of GW harmonics (and their phase velocity spectrum), thus linking

the GW sources to the meteorology of the lower atmosphere (*Yiğit et al.*, 2009; *Medvedev et al.*, 2011b). This launch level is around 260 Pa ($\sim 8$ km).

In the simulations to be presented, the vertical fluxes due to subgrid-scale GWs (1) are computed in all grid points in a time-dependent fashion for varying atmospheric conditions. These fluxes are used for calculating GW dynamical effects, i.e.,
GW-induced momentum deposition ("drag") and GW thermal effects, i.e., heating/cooling rates (*Yiğit and Medvedev*, 2009; *Medvedev and Yiğit*, 2012), which are interactively fed into the MGCM. In the absence of dissipation ($\tau = 1$), momentum fluxes per unit volume $\rho\overline{\mathbf{u}'w'}$ remain constant, and GWs do not affect the large-scale wind and temperature fields, that is, the large-scale fields that are self-consistently resolved by the MGCM. If $\tau$ falls below unity due to dissipative effects, then GWs influence the atmospheric circulation and thermal structure. This behavior represents the process in which GWs interact with
the background flow continuously as they propagate upward. This implementation also alleviates the limitation of the linear breaking assumption assumed by the majority of the conventional GW parameterizations. In a realistic atmosphere GW inter-actions with the background atmosphere is continuous and occurs in a nonlinear fashion. The rate of GW dissipation/breaking, which itself depends on the simulated flow, determines the momentum and thermal forcing.

## 2.3   Linking Gravity Waves and Ice Clouds.

As was described above, the GW parameterization calculates covariances of wave field variables. Of particular interest is the amplitude of temperature fluctuations $|T'| = \sqrt{\overline{T'^2}}$. Because this scheme does not provide phase information about the subgrid-scale GW field, instantaneous values of the parameterized (unresolved by the model) temperature disturbances $T'$ are impossible to determine. However, $|T'|$ quantitatively characterizes possible maxima of fluctuations in a given point, thus allowing for extending the probabilistic approach to $CO_2$ cloud formation. We assume that a cloud can form, if the total
temperature $T - |T'|$ drops below a certain threshold $T_s$. Then, we define the probability $P$ of this event as

$$P(z) = \begin{cases} 1 & \text{if } T - |T'| \leq T_s \\ 0 & \text{otherwise.} \end{cases} \tag{2}$$

In the paper, we loosely call it the "probability of $CO_2$ cloud formation". In fact, cold temperature is a necessary, but not the sufficient condition for clouds to form. Microphysics of condensation is more complex and involves an existence and characteristics of nuclei particles. Therefore, $P$ must be treated as a certain metric introduced for quantifying conditions favoring formation of a cloud. Because of the probabilistic nature of $|T'|$ itself, $P$ has a meaning only after a certain averaging.
For example, calculating $P$ at every model time step within a certain time interval and dividing by the number of the step yields the probability $0 \leq \bar{P} \leq 1$ as a percentage of time when cloud formation was possible.

To determine $T_s$, we consider the Clausius-Clapeyron equation that relates pressure $p$ and temperature $T$ in a system con-sisting of two phases, as is the case for carbon dioxide ($CO_2$) on Mars

$$\left(\frac{dp}{dT}\right)_{sv} = \frac{L_{sv}}{T(\nu_v - \nu_s)}, \tag{3}$$

where $L_{sv}$ is the latent heat of sublimation (the subscripts $s$ and $v$ denote the conversion from solid to vapor phases), $\nu_v$, and
$\nu_s$ are the specific volumes for vapor and solid phases, respectively. Since $L_{sv}$ is the heat input into the system and, thus, is

positive, $\nu_v \gg \nu_s$, the sublimation pressure curve is always positive and the latent heat is temperature independent. Thus, the vapor phase of carbon dioxide behaves like an ideal gas and the Clausius-Clapeyron equation can be integrated to obtain the expression for the saturation temperature $T_s$

$$T_s = \left\{ \frac{1}{T_0} - \frac{R\ln[p(z)/p_0]}{L_{sv}} \right\}^{-1}, \tag{4}$$

where $T_0 = 136.3$ K is the reference saturation temperature at $p_0 = 100$ Pa and $L_{sv} = 5.9 \times 10^5$ J kg$^{-1}$. As suggested by previous experimental constraints (*Glandorf et al.*, 2002) a significant degree of supersaturation is required, if microphysics of condensation is accounted for. We employ for the saturation pressure the value $1.35 \times p$ instead of $p$ in Equation (4). This estimate corresponds to nuclei particles with sizes bigger than 0.5 $\mu$m and was used in previous MGCM studies (e.g., *Colaprete et al.*, 2008; *Kuroda et al.*, 2013). The same supersaturation threshold is applied in the condensation/sublimation scheme utilized by the MPI-MGCM for explicitly accounting for resolved $CO_2$ phase transitions. For smaller nuclei particles, which are expected to be present in the upper atmosphere, the degree of supersaturation increases.

## 2.4 Martian General Circulation Model Simulations

After a multi-year spinup, the model was run for a full Martian year (669 sols $\sim$ 687 Earth days) under the low-dust scenario and for low solar activity conditions. The dust scenario represents a composite of measurements by the Thermal Emission Spectrometer onboard Mars Global Surveyor (MGS-TES) and the Planetary Fourier Spectrometer onboard Mars Express (MEX-PFS) with the global dust storms removed. Two full-Martian-year experiments have been performed: without GWs included (EXP0) and with the GW scheme turned on (EXP1). The results to be presented are based on daily averaged output data.

## 3 Mean Fields, Gravity Waves, and Probability of $CO_2$ Ice Cloud Formation at Solstice and Equinox

Gravity waves can facilitate $CO_2$ cloud formation in two ways: a) by cooling down the large-scale atmosphere globally, thus bringing its temperature closer to the condensation threshold, and b) by locally creating pockets of cold air. In this section, we explore the former effect by comparing the EXP0 (no-GW run) with EXP1 (GW-run) simulations. It is instructive to compare the effects produced by GWs with the other major cooling mechanism in the middle and upper atmosphere of Mars, – cooling due to radiative transfer in the IR $CO_2$ bands. A detailed study of the two mechanisms using two Martian GCMs has been performed for a vernal equinox (*Medvedev et al.*, 2015). Here our emphasis is on the global and seasonal effects.

**Figure** 1 presents the annual global means of the simulated temperature ($T$), GW-induced thermal heating-cooling rates ($Q_{GW}$), and of $CO_2$ radiative cooling rates ($Q_{CO_2}$) for the experiments EXP0 ( dashed line) and EXP1 (solid line). The left panel demonstrates that inclusion of GW effects cools down the upper atmosphere at all altitudes above 60 km in a global sense, e.g., the temperature in the mesosphere above 100 km is by $\sim$10 K lower. Note that this change includes both thermal and dynamical influence of GWs. The thermal one is due to GW-induced heating/cooling rates, while the dynamical channel encompasses the temperature field response to acceleration/deceleration of the large-scale wind by small-scale GWs. Here, it

is not our goal to explore the two channels in more detail. More importantly within the context of this paper is to demonstrate the appreciable net cooling effect of GWs.

Figure 1b shows that $CO_2$ cooling is present at nearly all altitudes in the middle atmosphere, and peaks with more than –80 K sol$^{-1}$ around 90 km, steeply decreasing above. On the contrary, GW cooling rates increase with altitude, exceeding that of
$CO_2$ in the upper mesosphere and lower thermosphere, and peak with –80 K sol$^{-1}$ at 140 km. Around the mesopause and lower thermosphere, GWs cool down the atmosphere by $\sim 5 - 8\%$ (Figure 1c). It is also seen that the GW-induced effects modulate the $CO_2$ cooling via changes in the background temperature: $CO_2$ cooling is about up to 60% weaker in the run with GWs. In the rest of the paper, we present the results of simulations that include GW effects (EXP1).

**Figure** 2 illustrates the altitude-latitude distributions of the zonal mean temperature and wind for two characteristic seasons:
the vernal equinox (averaged over 42 sols corresponding to $L_s = 0° - 20°$, left panels) and for the aphelion solstice (44-sol average, $L_s = 90° - 110°$, right panels). The simulated temperatures below $\sim$70–80 km are in a good agreement with observations, where systematic satellite measurements are available (e.g., *Smith*, 2008). The coldest temperatures on Mars (favoring $CO_2$ condensation) are near the mesopause. During the equinox, the minimum of 120 K is over the equator. At the aphelion season, the mesopause is colder and the temperature minimum shifts to the summer hemisphere. This behavior is
tightly related to the wind distributions. It is seen that, in both seasons, zonal jets reverse their directions near the mesopause. The similar phenomenon is well known in the mesosphere and lower thermosphere of Earth, and is caused by the deposition of zonal momentum (i.e., zonal drag) by GWs of lower atmospheric origin (of up to –250 m s$^{-1}$ sol$^{-1}$ in this case), as demonstrated by the black contour lines. During the northern hemisphere summer solstice, the asymmetry between the two hemispheres is significant. Easterly and westerly jets dominate in the northern summer and southern winter hemispheres,
correspondingly, with the middle atmospheric jets extending higher up and reversing their directions between 110 and 120 km due to zonal GW drag acting against the mean winds. The zonal mean drag increases from $\pm 50$ m s$^{-1}$ sol$^{-1}$ to $\pm \sim 1000$ m s$^{-1}$ sol$^{-1}$ from the mesosphere to the lower thermosphere, with relatively asymmetric distribution between hemispheres. The drag of similar magnitudes has been inferred from aerobraking data in the Martian lower atmosphere (*Fritts et al.*, 2006).

**Figure** 3 demonstrates that the probability $P$ of $CO_2$ cloud formation is strongly determined by the mean temperature. It
is seen that the saturation condition for clouds are more likely to be met during the solstice than the equinox. Specifically, the cloud formation can occur in $\sim$1% of the time in the equatorial mesosphere during the equinox. Higher up at around 120 km, the cloud formation probability increases and reaches 4.5% with larger values found in the northern high-latitudes. During the solstice, the probability $P$ is larger in all atmospheric regions. In particular, a very strong cloud formation is seen in the winter polar lower atmosphere (Figure 3b), which, however, is not the focus of this paper. In the upper atmosphere, the peak values of
$P$ exceed 30% between 100 and 120 km in the northern hemisphere at middle- and high-latitudes. A closer examination of Figures 2 and 3 reveals that the distributions of $P$ and mean temperature are not identical. GW-induced fluctuations $|T'|$, which are a measure of GW activity, also contribute to $CO_2$ supersaturation, especially in low- to middle-latitudes of the middle atmosphere in both seasons and in high-altitude polar regions. Overall, large GW-induced temperature fluctuations prevail above 100 km up to 140 km, primarily located over the equator and in high-latitudes of both hemispheres.

## 4 Seasonal Variation of the Mean Fields

We next investigate the seasonal variations of the simulated temperature and wind in more detail by focusing on three representative altitudes in the mesosphere and lower thermosphere: 80, 100 and 120 km. There are too few observations at these altitudes to date to validate the simulations. The exception is the temperature at $\sim 80$ km (**Figure** 4a), which can be directly compared to retrievals from Mars Climate Sounder (MCS) onboard Mars Reconnaissance Orbiter (MRO) (see Figure 10 in the paper of *Sefton-Nash et al.*, 2013). Both observations and simulations demonstrate a relatively symmetric with respect to the equator distributions during equinoxes ($L_s = 0°, L_s = 180°$). The lowest and highest temperatures occur in the southern hemisphere during winters and summers, correspondingly. The model generally reproduces the observed temperature well, except that it overestimates it in the southern hemisphere winter by up to 20 K. Alternating with seasons zonal winds at $\sim$80 km represent an extension of the lower- and middle atmosphere jets formed as a consequence of the Coriolis force acting on the summer-to-winter meridional circulation cell.

In the upper mesosphere (100 km, Figure 4b,e) the simulated temperature and wind show variations similar to that at 80 km, but with noticeably colder temperatures. Around the mesopause (120 km, Figure 4c,f), the simulated seasonal variations differ significantly from those in the mesosphere. It is seen that the coldest temperatures of down to 90–100 K are found around the summer high-latitudes at solstices, and the temperature distributions are hemispherically less symmetric. The summer high-latitude hemispheres are remarkably different. Polar temperatures fall down to 90 K in the summer hemisphere at the aphelion and to 115 K at the perihelion. The zonal winds reverse their directions at 120 km, which is especially well seen during the aphelion season. During other seasons, the simulated winds demonstrate a significant weakening as compared to distributions in the mesosphere. The latter is primarily attributed to the GW drag, which we present next along with GW-induced temperature fluctuations.

## 5 Seasonal Variation of Gravity Wave Activity and Probability of CO$_2$ Ice Clouds

Parameterized GW-induced temperature fluctuations ($|T'|$), GW drag ($a_x$), and probability of cloud formation ($P$) are studied next in Figure 5 in the same manner as temperature and zonal winds are presented in Figure 4. Overall, the seasonal variations of the parameterized GW-induced temperature fluctuations, which are created by GW harmonics that survived propagation from the lower atmosphere depend on the assumed wave sources and on filtering by the underlying mean winds. In the mesosphere (80 km, Figure 5a), the fluctuations of up to 16 K enhance at middle- to high-latitudes and during the solstices with slightly larger magnitudes during winters. The middle column of Figure 5 shows the seasonal variations of the zonal GW drag, which is largely determined by the background winds below presented in Figure 4 and characterizes the rate of change of GW momentum fluxes with height. It is seen that it is directed mainly against the mean flow throughout the mesosphere. Finally, the probability $P$ of cloud formation is plotted in the rightmost column of Figure 5. A continuous presence of $P$ of up to 2-4% is seen around the equator at 80 km nearly throughout the entire Martian year. After the northern summer hemisphere solstice (aphelion), regions of cloud formation gradually expand to lower-latitudes ($\pm 30°$), resembling a fork-like structure, in some level of agreement with *Sefton-Nash et al.* (2013)'s observations. During southern winter solstice , the probability of cold

pocket formation is somewhat present around midlatitudes. There is some degree of correlation between the cloud formation probability and GW activity represented as fluctuations and drag.

In the upper mesosphere (100 km, middle row), GW-induced temperature fluctuations increase, along with the GW drag imposed on the mean circulation, and the cloud formation probability demonstrates a more definitive correlation with the GW activity during all seasons. Cold pockets occur more frequently at middle- and high-latitudes ($P \sim 16 - 20\%$), exceeding the equatorial cloud probability rate. Around the mesopause, GW-induced fluctuations increase further maximizing at middle- and high-latitudes with values of up to 26 K during both aphelion and perihelion. The probability $P$ increases to more than $\sim 30\%$ correspondingly.

## 6 Discussion

As mentioned in the description of the model, the $CO_2$ condensation/sublimation scheme employed in the MPI-MGCM is able to resolve $CO_2$ ice formation and annihilation when temperature in a grid point crosses the condensation threshold. In our simulations, there were very few occurences of such clouds in the mesosphere above 60 km to offer a reliable statistics. Inclusion of GW effects leads, generally, to colder simulated temperatures, which provide favorable conditions for cloud formation. This cooling in the mesosphere is mainly produced via the dynamical channel due to GW-induced changes in the winds that affect temperature through the thermal wind relation, rather than via the thermal channel due to direct heating/cooling by dissipating GW harmonics. The latter clearly transpires in experiments with thermal effects of the parameterized waves turned on and off (not shown). The direct thermal effects of GWs increasingly grow with height and become important near the mesopause and above.

The vast majority of studies report on cloud observations in the Martian mesosphere below $\sim 80$ km. *Sefton-Nash et al.* (2013) analyzed data from Mars Climate Sounder (MCS) on board the Mars Reconnaissance Orbiter (*Graf et al.*, 2005; *Zurek and Smrekar*, 2007) (MRO) during dayside and nightside local times over two Martian years and provided a global picture of high-altitude   clouds. They found out that the distribution of clouds over latitude and season does not appear to vary between each Martian year and that clouds occurred more often   in low latitudes during the aphelion season and concentrated around two midlatitude bands during perihelion. Using two different observational modes, *Sefton-Nash et al.* (2013) showed that the latitudinal distributions of clouds varied little between the different local times in the second half of the year. It must be noted that *Sefton-Nash et al.* (2013) could not discriminate between $CO_2$ and water clouds. The majority of positively identified $CO_2$ cloud observations took place in the first half of the year with only a few detections in the second half. On the other hand, water ice clouds usually do not extend higher than $\sim 40$ km except during perihelion, when they rise to 60-65 km. Therefore, all clouds observed above 70 km are likely not water ice clouds. The question regarding the nature of these detected clouds is still open. Our simulations illustrate that favorable conditions for $CO_2$ condenstation in the mesosphere exist in low latitudes throughout the year (Figure 5g). This is likely because the mean temperature is lowest around the equator at all seasons (Figure 4a), while GW-induced temperature fluctuations are contrary small (Figure 5a). The model also predicts a higher probability of cloud formation in midlatitudes of the summer hemisphere during both solstices.

It is observationally challenging to determine the precise altitude of $CO_2$ ice clouds and there are intrinsic limitations of the retrieval algorithms associated with $CO_2$ cross-sections (*Määttänen et al.*, 2013). Nevertheless, our modeling results can qualitatively be compared with *Sefton-Nash et al.* (2013)'s analysis of the seasonal variation of Martian high-altitude clouds. The observations show that during northern hemisphere summer, clouds formed in the mesosphere more rarely than during

perihelion and located mainly around the equator. Previous analysis of the data from the Thermal Emission Spectrometer (TES) on board the Mars Global Surveyor (MGS) (*Clancy et al.*, 2007) also indicated that cloud occurrences were confined to a narrow latitude sector of $\pm 15°$ during the aphelion season ($L_s = 30° - 150°$). In agreement with observations, our simulations show higher probabilities of cloud formation in low latitudes throughout all seasons (Figure 5g). The latter is simply a consequence of the temperature minimum near the equatorial mesopause. The model reproduces more favorable conditions

for $CO_2$ condensation in the midlatitude regions during wintertime. It agrees with observations in that mesospheric clouds occur more frequently during perihelion (Figure 5g). Cold pockets with supersaturated temperatures occur in the model only in less than 10% of time at $\sim$80 km, but their probabilities grow with height. At 100 km, maxima of $P$ of up to $\sim$18% are seen in midlatitudes of the summer hemispheres (Figure 5h), while higher up at $\sim$120 km these maxima exceed $\sim$30% and shift poleward (Figure 5i). Such behavior is due to growing with height amplitudes of GWs and the associated temperature

fluctuations (Figure 5a-c). The shift of maxima of cloud probabilities first to middle and then to high latitudes is caused by the cold anomaly of the mean temperature, which is induced by GWs in the mesosphere. The similar GW-induced cold summer mesopause anomaly is well known in the atmosphere of Earth (*Garcia and Solomon*, 1985).

There are certain disagreements between the modeled cloud formation probabilities and existing observations at high altitudes (80 km and above). In particular, the day-side observations of *Sefton-Nash et al.* (2013) demonstrate a more symmetric

with respect to the equator distribution of $CO_2$ clouds during the first part of the year. They do not show a "pause" near the equinox around $L_s = 180°$, which is clearly seen in Figures 5h,f. It is worth noting that the superposition of the simulated patterns at 80 and 100 km is close to the superposition of the observed night- and day patterns (*Sefton-Nash et al.*, 2013, Fig. 6) attributed to the 80 km altitude. Finally, there is no statistically significant observational support for the predicted cloud formation probability above 80 km. We, therefore, discuss possible reasons and shortcomings of the modeling methodology.

One source of uncertainty in our simulations is the assumed degree of supersaturation, which is currently 35% based on previous experimental constraints (*Glandorf et al.*, 2002). However, a variable with height supersaturation threshold is possible, which could modulate $P$ in our numerical experiments. This variable threshold may reflect the microphysics of cloud formation, which implies an existence of nuclei and strong dependence on their sizes. It is likely that the existence of cold pockets (the necessary condition for cloud formation) is far from sufficient for clouds to form, especially in the upper atmosphere. Thus,

the lack of nuclei in in the upper atmosphere may prevent cloud formation. If formed, ice particles must be small (being of submicron size) and clouds are too thin to be previously detected.

In all our simulations, only probabilities of GW-induced clouds were calculated and, thus, no radiative effect of such clouds were taken into account. Such radiative feedback has been considered, for example, in the work by *Siskind and Stevens* (2006) in the Earth context. However, one can expect that, given that IR $CO_2$ and GW thermal effects together dominate the energy

budget of the mesosphere (e.g., *Medvedev et al.*, 2015), secondary radiative processes are likely to play a relatively minor role

in the cloud formation by producing local modulations of temperature. A more comprehensive examination of the radiative feedback processes in the Martian environment would require a two-way coupling between microphysics and small- and large-scale dynamics. Interestingly, using a one-dimensional radiative-convective model, *Mischna et al.* (2000) demonstrated that the lower atmospheric $CO_2$ clouds have a potential to produce an additional cooling of the Martian surface by reflecting the incoming solar radiation. A further source of uncertainty is the use of one-dimensional atomic oxygen profile, which may affect neutral temperatures.

An obvious candidate for explaining mismatches between the modeling and observations is the specification of sources in the GW parameterization. In the simulations, we assumed a globally uniform and constant with time distribution of GW momentum fluxes. The magnitudes of the fluxes were chosen from observations to capture the "background" effect of small-scale waves, as described in detail in our earlier works (*Medvedev et al.*, 2011b, a). Recently, using a high-resolution Martian GCM, *Kuroda et al.* (2015, 2016) have shown a strong seasonal and latitudinal variation of GW momentum fluxes in the lower atmosphere and, as a result, significant variations of GW-induced activity in the middle atmosphere. Constraining wave sources is a logical next step in model development, which can potentially improve simulations of clouds.

Finally, other limitations in the model can result in imperfections with the simulated mean fields and, as a consequence, with erroneous estimates of cloud formation probability $P$. The Reviewer suggested that accounting for radiative effects of water clouds and for more realistic dust scenario (mainly associated with its vertical distribution) may affect the simulated $P$. These and other undertaken paths of MGCM sophistication, like self-consistent modeling of water and aerosol cycles, can potentially bring observations and simualations of $CO_2$ clouds closer.

## 7  Summary and Conclusions

We presented simulations with the Max Planck Institute Martian General Circulation Model (MPI-MGCM) (*Medvedev et al.*, 2013), incorporating a whole atmosphere subgrid-scale gravity wave (GW) parameterization of *Yiğit et al.* (2008), of distributions of mean fields, GW effects, and cloud formation probabilities over one Martian year, assuming multi-year averaged observed dust distribution with major dust storms removed. Model results are compared to a run without subgrid-scale  effect included.

Inclusion of effects of small-scale GWs facilitates $CO_2$ cloud formation in two ways. First, they cool down the upper atmosphere globally and, second, they create excursions of temperature well below the $CO_2$ condensation threshold in some parts of the middle atmosphere. The main findings of this study are as follows.

1. GWs lead to ∼9% colder global annual mean temperatures and even stronger temperature drops locally. Global annual mean GW-induced cooling of $-30\,\mathrm{K}\,\mathrm{sol}^{-1}$ is comparable with that of due to radiative transfer by $CO_2$ molecules around 100 km and exceeds it above, reaching $-80\,\mathrm{K}\,\mathrm{sol}^{-1}$ around 140 km. GW-induced effects modulate the $CO_2$ cooling via changes in the background temperature.

2. Simulations reveal strong seasonal variations of GW effects in the upper mesosphere and lower thermosphere with solsticial maxima: Eastward GW drag peaks during the summer solstices and westward GW drag maximizes around the winter solstices with up to $\pm 1000$ m s$^{-1}$ sol$^{-1}$.

3. Around the mesopause, GW-induced temperature fluctuations $|T'|$ can exceed 20 K and the ice cloud formation probability ($P$) can be greater than $20\%$ locally.

4. Overall, GW temperature fluctuations substantially correlate with the cloud formation probability, in particular at middle- and high-latitudes in the upper mesosphere and mesopause region during all seasons.

5. Cloud formation exhibits strong seasonal variations larger than $30\%$, with summer solsticial maxima at high-latitudes in the mesosphere and around the mesospause.

6. The simulated seasonal variations of clouds probabilities in the mesosphere are in a reasonable agreement with previous detections of two distinct mesospheric types of clouds, i.e., equatorial and midlatitudes clouds.

This study has shown that accounting for GW-induced temperature fluctuations in the Martian GCM reproduces supersaturated cold temperatures in the upper mesosphere throughout all seasons. GWs maintain globally cooler air, which is necessary for ice cloud formation, and help to explain some features of the observed seasonal behavior of high-altitude $CO_2$ ice clouds. Owing to GW-induced globally colder temperatures and local temperature fluctuations, high-altitude clouds can form from the upper mesosphere to the mesopause region, and occasionally even slightly above the mesopause. We conclude that GW dynamical and thermal effects not only maintain the colder Martian mesosphere and lower thermosphere, but also significantly contribute to the specific features of the observed high-altitude clouds and their seasonal variations.

This study also puts forward new questions. Are our results concerning shaping the seasonal behavior of ice clouds model-specific? Can $CO_2$ clouds form at altitudes above the mesopause, as the simulations predict? How does the microphysics of cloud formation modify these predictions? Further systematic modeling and obervational efforts have to be performed in order to address these open questions.

*Data availability.* Upon request, the data used for the publication of this research is available from EY (eyigit@gmu.edu)

## Appendix A: Martian parameters and seasons

Mars demonstrates in terms of planetary parameters some similarities as well as differences to Earth as summarized in Table A1.

**Table A1.** Some key planetary parameters of Earth and Mars.

| Planetary parameters | Mars | Earth |
|---|---|---|
| Mean solar distance [AU] | 1.52 | 1 |
| Radius [km] | 3389 | 6370 |
| Length of a day [h] | 24.65 | 24 |
| Length of year [days] | 687 | 365.5 |
| Axial tilt [degrees] | 25.19° | 23.5° |
| Gravity [m s$^{-2}$] | 3.72 | 9.81 |
| Eccentricity | 0.0934 | 0.0167 |

The solar distance designates the average distance from Sun, given in terms of AU $\sim$ 150 million km. The axial tilt, or obliquity of the orbit, is measured with respect to the orbital plane. One solar day on Mars is referred to as one sol.

In planetary atmospheres, one "sol" refers to the duration of a solar day on Mars. The length of a day is longer on Mars than on Earth. One Martian sol is about 24 hours and 39 minutes (i.e., 24.65), thus slightly longer than and an Earth day. One Martian year is 687 days long or 669 Martian sols. Due to different eccentricities of Mars and Earth, their distance can vary significantly over the course of their orbital motion around Sun. Martian seasons are described by the solar longitude $L_s$. In our modeling, by convention, $L_s = 0$ vernal equinox, $L_s = 90°$ is northern hemisphere solstice (aphelion), $L_s = 180°$ is autumnal equinox, and $L_s = 270°$ is northern hemisphere winter solstice (perihelion).

*Competing interests.* The authors declare that they have no conflict of interest.

*Acknowledgements.* The modeling data supporting the figures presented in this paper can be obtained from EY (eyigit@gmu.edu). This work was partially supported by German Science Foundation (DFG) Grant HA3261/8-1 . EY was funded by the National Science Foundation (NSF) grant AGS 1452137.

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

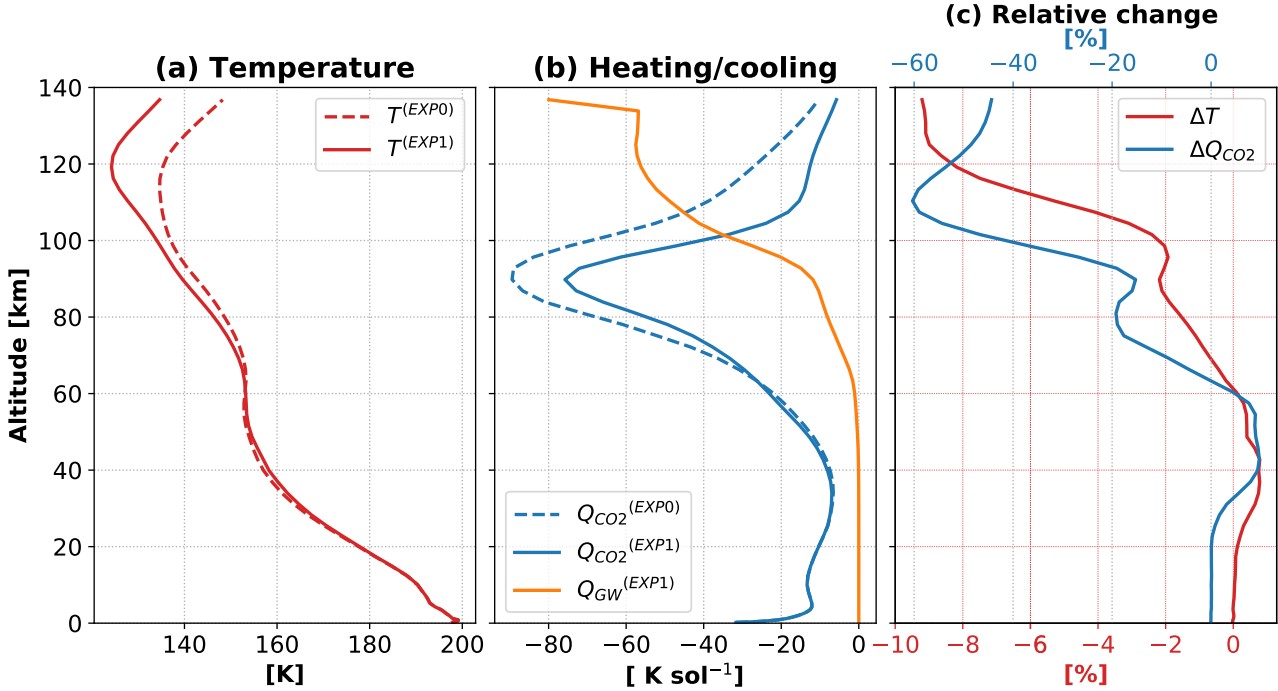

**Figure 1. Global annual mean temperature, and cooling by gravity waves and radiative processes in carbon dioxide molecules**. **(a)** Globally averaged annual mean neutral temperature ($T$ [K]) with gravity waves (solid line, "EXP1") and without gravity waves (dashed lines, "EXP0"); **(b)** Globally averaged annual mean gravity wave heating/cooling $Q_{GW}$ [K sol$^{-1}$] (orange line) and $CO_2$ 15 $\mu$m cooling ( blue line); **(c)** Relative percentage change with respect to EXP0 simulations for the temperature (red) and $CO_2$ cooling ( blue), calculated as $[T^{(EXP1)} - T^{(EXP0)}]/T^{(EXP0)}$ and $[Q_{CO2}^{(EXP1)} - Q_{CO2}^{(EXP0)}]/Q_{CO2}^{(EXP0)}$. In both panels dashed lines represent the simulation without gravity waves effects, while the solid lines are for the simulation with gravity wave propagation from the lower atmosphere upward. The annual mean refers to an averaging over one Martian year (669 sols = 687 Earth days) over all longitudes and latitudes.

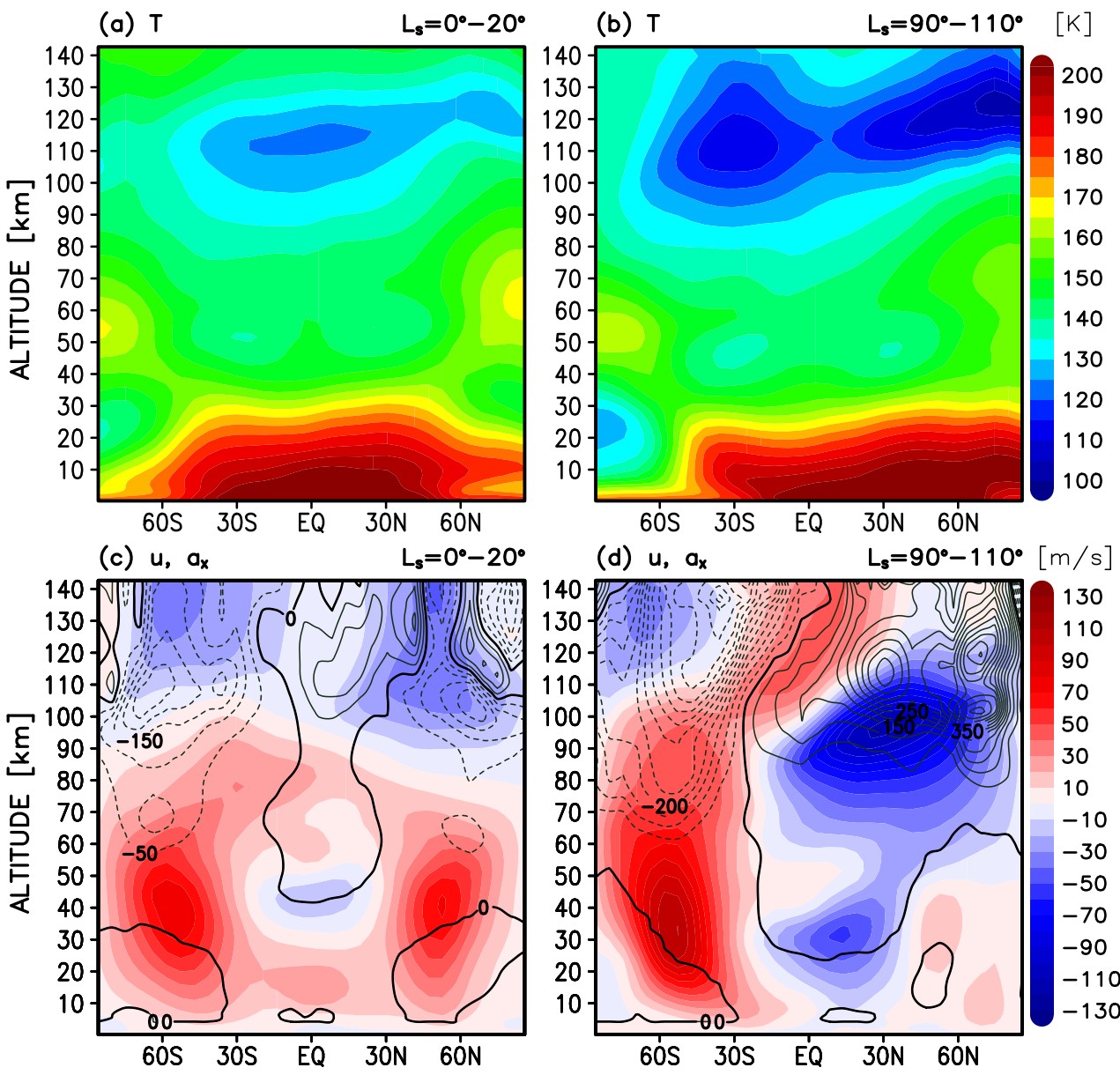

**Figure 2.** Altitude-latitude distributions of the mean zonal mean fields during vernal equinox and northern summer solstice (aphelion): **(a)** temperature $(T)$ at equinox, **(b)** temperature $(T)$ at solstice **(c)** zonal wind $u$ (color shaded) and zonal GW drag (acceleration/deceleration) $a_x$ (contour lines) **(d)** zonal wind $u$ (color shaded) and zonal GW drag $a_x$ (contour lines). The fields are averaged over $L_s = 0° - 20°$ (42 sols) for vernal equinox and over $L_s = 90° - 110°$ (44 sols) for northern hemisphere summer period. Temperature is in units of K, the zonal wind is in m s$^{-1}$, and the zonal GW drag is in m s$^{-1}$ sol$^{-1}$. Red and blue shading in the zonal wind plot represent the easterly (westward) and westerly (eastward) wind systems. Dashed and solid lines for the drag are for the easterly and westerly wave drag in intervals of 50 m s$^{-1}$ sol$^{-1}$.

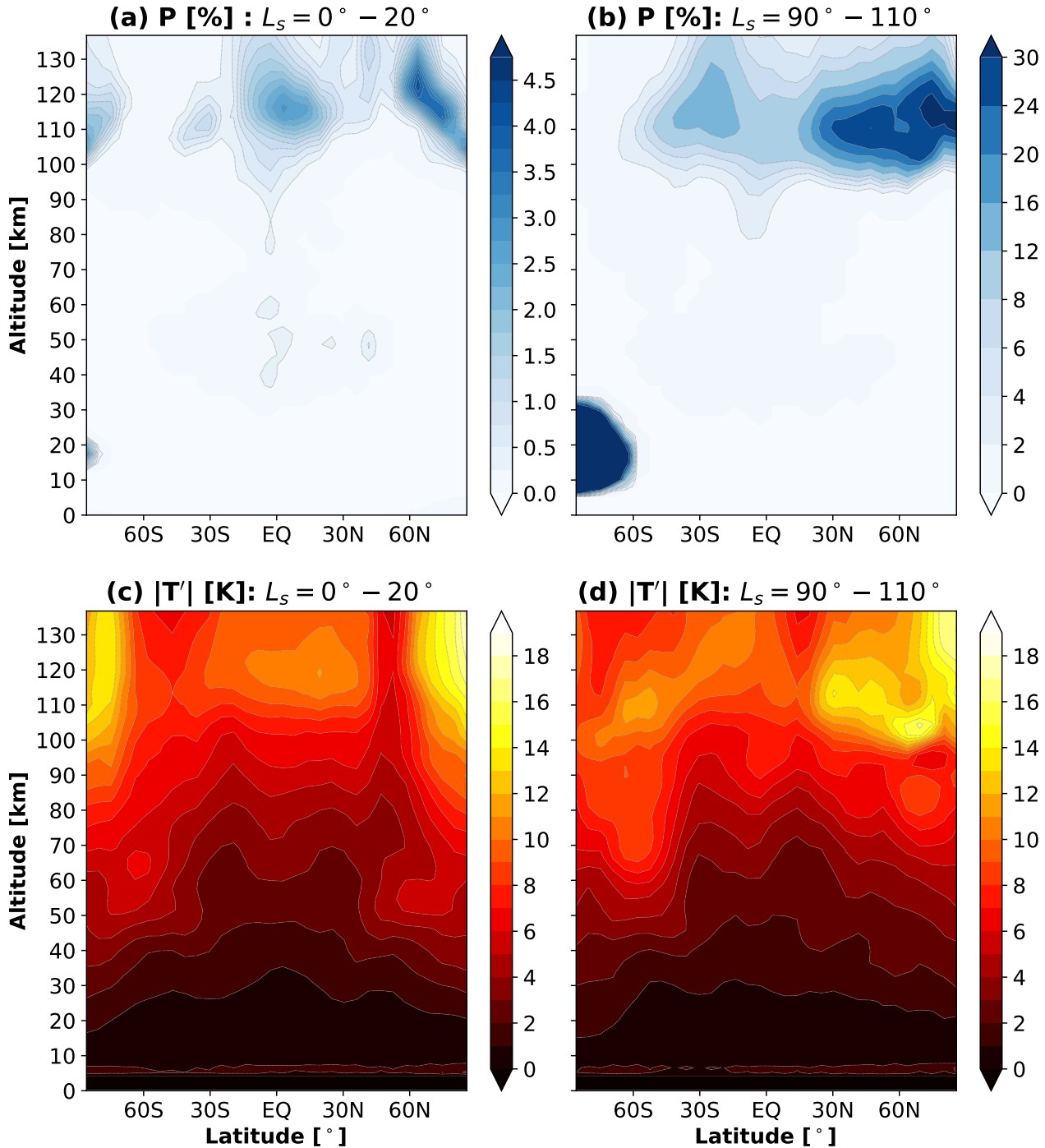

**Figure 3. Altitude-latitude distributions of mean zonal mean cloud probability and gravity wave effects:** (a) cloud probability (P) at equinox, (b) cloud probability at solstice, (c) gravity wave induced temperature fluctuations ($|T'| = T'_{gw}$) at equinox, (d) gravity wave induced temperature fluctuations at solstice. The fields are averaged over a period of $L_s = 0° - 20°$ (42 sols) for vernal equinox and over $L_s = 90° - 110°$ (44 sols) for northern hemisphere summer solstice seasons, i.e., in the same manner as the data presented in **Figure** 2. Temperature fluctuations are in units of K and the probability is expressed in terms of percentage.

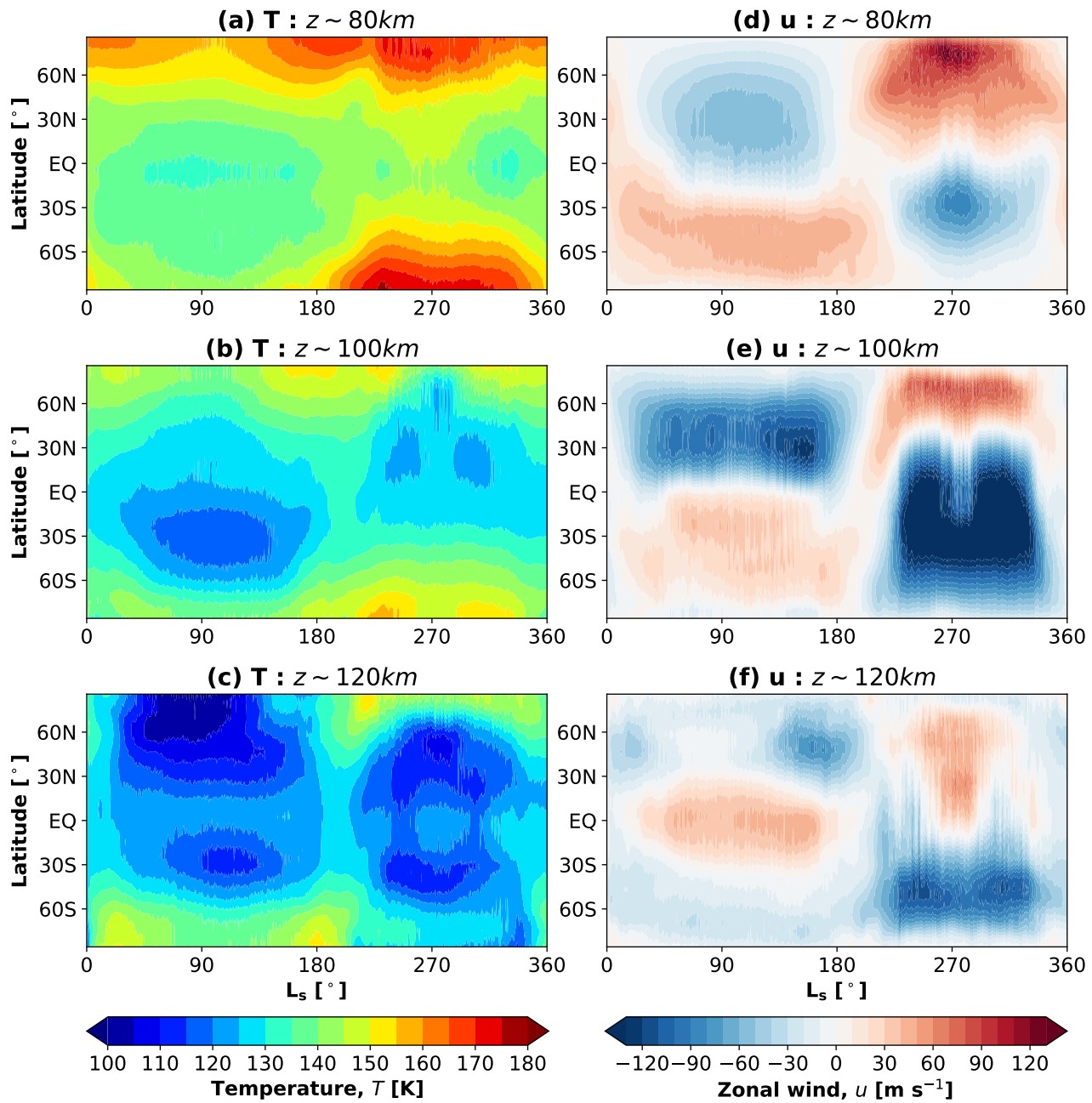

**Figure 4. Seasonal variations of mean (i.e., daily and zonally averaged) atmospheric fields: (a)** Temperature ($T$) at 80 km, **(b)** $T$ at 100 km, **(c)** $T$ at 120 km, **(d)** Zonal wind ($u$) at 80 km, **(e)** $u$ at 100 km, **(f)** $u$ at 120 km. Temperature is in K and the zonal wind is in m s$^{-1}$. Red/blue shading for the wind represent eastward/westward winds. A Martian year has about 669 sols , which is plotted in terms of solar longitude $L_s$ (in degrees) from $L_s = 0° - 360°$. . $L_s = 0°$ marks the vernal equinox in the northern hemisphere. $L_s = 90°$ and $L_s = 270°$ are aphelion and perihelion seasons, respectively.

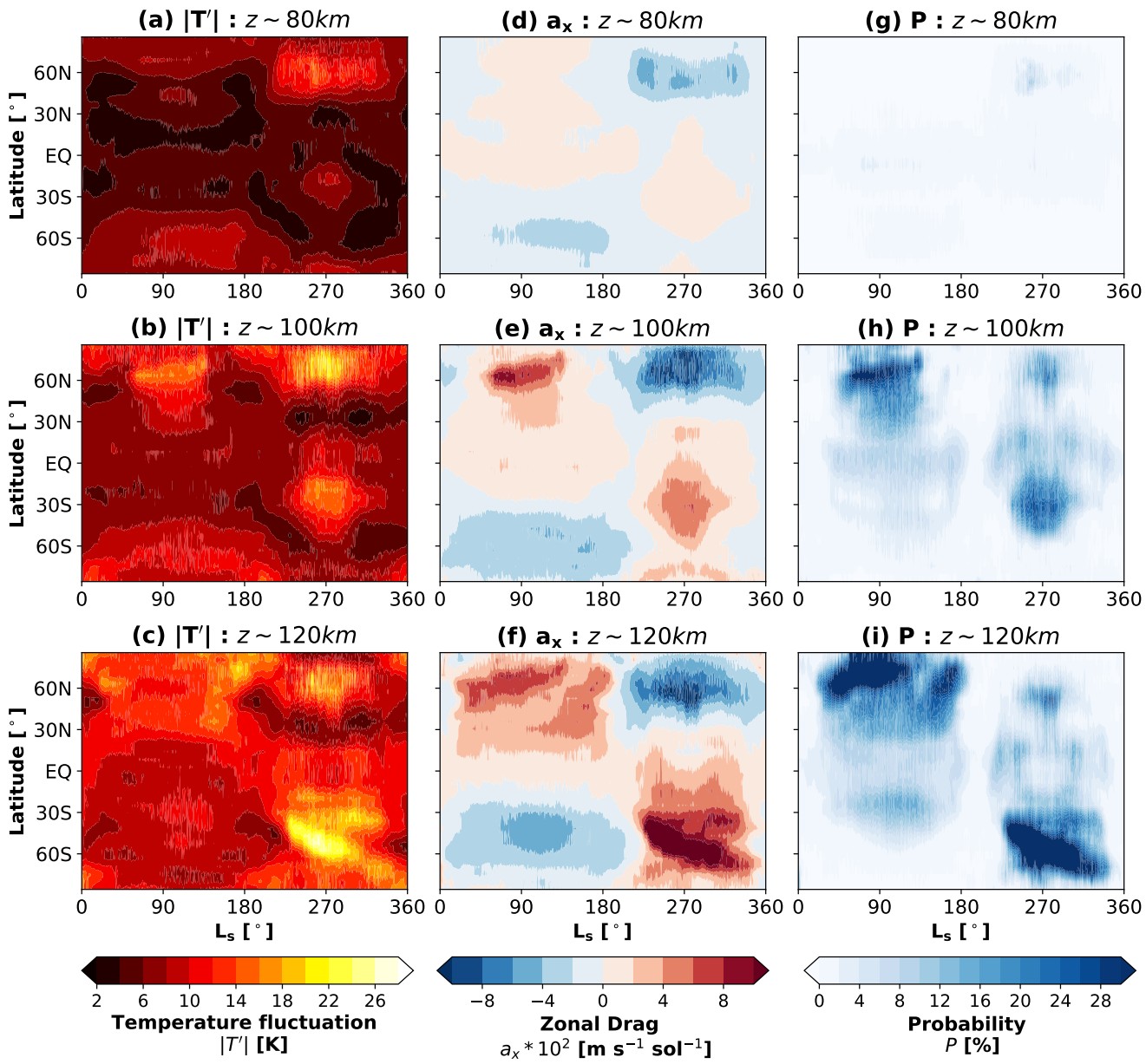

**Figure 5. Seasonal variations of cloud formation probability, GW drag and GW-induced temperature fluctuations: (a)** GW-induced temperature fluctuations ($|T'|$) at 80 km, **(b)** $|T'|$ at 100 km, and **(c)** $|T'|$ at 120 km, **(d)** GW zonal drag ($a_x$) at 80 km, **(e)** $a_x$ at 100 km, **(f)** $a_x$ at 120 km, **(g)** Cloud formation probability ($P$) at 80 km, **(h)** $P$ at 100 km, **(i)** $P$ at 120 km. Probabilities are in percentage; zonal drag is in m s$^{-1}$ sol$^{-1}$, and temperature fluctuations are in K. In the drag plots (**(d-f)**, red/blue represents eastward/westward GW drag. Presented model data are in terms of daily and zonal averages. Note that the zonal drag is plotted in 200 m s$^{-1}$ sol$^{-1}$ intervals. $L_s = 0°$ marks the vernal equinox in the northern hemisphere.