# Peer review of "Influence of gravity waves on the climatology of high-altitude Martian carbon dioxide ice clouds"

_Annales Geophysicae, 2018_

## Referee Comment (RC1) · Anonymous Referee #1 · 25 Jul 2018

**Review of the manuscript "Influence of gravity waves on the climatology of high-altitude Martian carbon dioxide ice clouds" by Yiğit et al., submitted to Annales Geophysicae**

**General comments**

The manuscript studies the links between the formation of CO2 clouds in the mesosphere of Mars and the effects of gravity waves propagating from the lower to the upper atmosphere. For this purpose, a General Circulation Model (GCM) incorporating a subgrid-scale parameterization of the effects of gravity waves, previously used to study the role of gravity waves on cloud formation for a single season, is run for a full Martian year. Gravity waves produce a global cooling of the upper atmosphere that facilitates the formation of clouds, and in addition induce local temperature perturbations producing cold air pockets. An statistical probability of cloud formation is derived from the model outputs, which the authors find to correlate well with the gravity wave activity in the model. The authors claim that the probability of cloud formation they derive is in good agreement with the observed seasonal and latitudinal climatology of mesospheric CO2 clouds.

The paper is well written and easy to read. The topic is of interest for the Martian community (and also for people studying high altitude clouds on other planets, including the Earth). The introduction adequately summarizes previous works on the topic and sets up the questions raised by them. The figures are appropriate, although some minor improvements could be made to facilitate the comparison with observations (see below). The length of the manuscript is also appropriate. However, I find the discussion about the comparison with the observations to be flawed due to the incorrect assumptions the authors make about the observations in Sefton-Nash et al. (2013) (see details below). This is an essential aspect of the manuscript that absolutely needs to be corrected before publication.

**Specific comments**

-In order to validate their predicted CO2 cloud formation probability, the authors chose to use the observations by the Mars Climate Sounder (MCS) instrument described in Sefton-Nash et al. (2013), who observed that mesospheric clouds were usually confined to a narrow latitudinal band around the equator during the first part of the Martian year, while during the second half of the year mesospheric clouds appeared in the mid latitudes of both hemispheres. The problem is that the authors are assuming that all observations in Sefton-Nash et al. (2013) are CO2 clouds, which is not the case. Most if not all of the aerosol layers observed by MCS in the mid latitudes during the second half of the year are not CO2, but H2O clouds or even dust, which has very different implications for the mesospheric temperatures involved. This is proven by MCS temperature measurements simultaneous to the cloud observations, showing that the mid-latitude clouds formed after Ls=150 occur at layers with atmospheric temperature generally >40K above the CO2 frost point. Quoting Sefton-Nash et al. (2013), pages 351-352 "Retrievals for features detected during dust storm season generally shows temperatures between 30 and 80 K higher than CO2 frost poing, suggesting that the formation of CO2 ice at mesospheric altitudes is far less likely during perihelion season". This is in contradiction with the manuscript, e.g. (page 9): " The model reproduces more favorable conditions for CO2 condensation in the midlatitude regions during wintertime. It agrees with observations in that mesospheric clouds occur

more frequently during perihelion (Figure 5g)". So, this last statement is not correct for $CO_2$ clouds, which are the ones relevant for this study.

In fact, all the observations of mesospheric clouds able to spectrally discriminate between $CO_2$ and $H_2O$ clouds (those made by the OMEGA/Mars Express, PFS/Mars Express and CRISM/MRO instruments) provide a very similar climatology of $CO_2$ mesospheric clouds, with almost all observations concentrating around the equator (and at restricted longitudinal corridors) for the Ls=0-60 and Ls=90-130 periods, with just a couple of clouds observed by OMEGA at latitudes around 50 in both hemispheres during winter (e.g. Aoki et al., 2018; Vincendon et al., 2011; Määttänen et al., 2010). THEMIS/Mars Oddyssey instrument observed a population of mesospheric clouds in the mid latitudes on the Northern hemisphere winter (McConnochie et al., 2010), but they could not discriminate between $CO_2$ and $H_2O$ clouds, and their low altitudes (~45-55 km) suggest they are more likely $H_2O$ clouds.

This observed climatology of unambiguous $CO_2$ mesospheric clouds presents significant differences with the model predictions summarized in Fig. 5. The most striking one is that the model predicts high cloud formation probability during the second half of the year both at the equator and at the mid-high latitudes of the Northern hemisphere (at 80 km) and at both hemispheres (100 and 120 km). However, $CO_2$ clouds have barely been observed during this period. Given the connection between the gravity wave activity and the predicted cloud formation, this would suggest that the predicted effects of gravity waves are too intense, at least in the mid-high latitudes during the second half of the year.

To summarize, the comparison between the observed cloud climatology and the model predictions should be based only on clouds spectroscopically unambiguously determined to be composed of $CO_2$. The differences between the observed and predicted $CO_2$ cloud climatology should be acknowledged, and the implications for the gravity wave activity in the model discussed.

-Another interesting aspect that deserves further discussion is the altitude variation of the cloud formation probability, predicted to be significantly larger at 120 km than at 80 km. Although the altitude of $CO_2$ mesospheric clouds is not easy to determine for most of the datasets, the current observational knowledge is that, at least during daytime, they are placed at altitudes of about 70-80 km (Schölten et al., 2010; Määttänen et al., 2010). During nighttime SPICAM has detected mesospheric clouds with altitudes around 100 km. No clouds have been detected, to my knowledge, around 120 km or higher, where the model predicts the higher cloud formation probability. I would like to see a discussion about this discrepancy in the manuscript.

-The GCM used in the study is very shortly described, apart from the gravity wave parameterization. While this is mostly OK given that the model has already been described in previous papers, I think the implementation of the physical processes affecting the temperatures in the mesosphere/lower thermosphere needs to be described to some extent. For example, what atomic oxygen distribution are you using among the different possibilities discussed in Medvedev et al., 2015?

-Page 4, lines 8-9. "In the middle and upper atmosphere of Mars, wave damping occurs due primarily to nonlinear wave-wave interactions (breaking and/or saturation) and molecular diffusion and thermal conduction, which are accounted for through the transmissivity". Eckermann et al., Icarus 211, pp. 429-442 (2011) showed that radiative damping can be a dominant process in the middle atmosphere of Mars. Do you consider it in your model?

-Figure 1. The gravity wave cooling is generally below 60 K/day except for the strong peak at 140 km reaching 120 K/day. Could you provide an the explanation for this strong peak? It apparently affects only one or two model layers, could you confirm this? Can it be due to any boundary effect? Note also that Fig. 1b) horizontal axis is labeled as K/day, but in the caption you state it is in units of K/sol, which is slightly different. Please correct.

-Page 8, lines 4-5: "The model generally reproduces the observed temperature well, except that it overestimates it in the southern hemisphere winter by up to 20 K". Other data-model discrepancies are evident by comparison of Fig. 4a with Figure 10 in Sefton-Nash et al., (2013). In particular, the temperature at 80 km in the polar regions can be higher than 180 K in MCS observations, while apparently (but this is maybe an artifact of the chosen color scale) do not go much higher than 150 K in the model. Could you please clarify it?

-Page 8, lines 10-11: "It is seen that the coldest temperatures of down to 90-100 K are found around the summer high-latitudes at solstices and during equinoxes". I do not see those low temperatures during equinoxes, when apparently temperatures do not go below 120 K, as can be seen also in Fig. 2a. Please clarify/correct.

-Page 9, lines 14-16. "Although the vast majority of studies report on cloud observations in the Martian mesosphere below ~80 km, there are some studies that extend their analysis to higher altitudes presenting detections of CO2 clouds at around the mesopause (~100 km) and above (e.g. Sefton-Nash et al., 2013)". Sefton-Nash et al. (2013) only detected clouds up to 90 km (e.g. their figure 9).

**Technical corrections**

-Page 2, line 1: "Because the Martian mesosphere is, in average, warmer ..." Warmer than the terrestrial one, or warmer than the CO2 frost point? Please specify.

-Page 3, line 5 "variations variations". Please remove one.

-Page 3, line 30: "It was developed in detail in the work of Yigit et al. (2008), the general principles of ...". I think either removing "in detail" or changing to "described in detail" would be more correct. Also please add "and" after the comma.

-Page 4, lines 29-30. "This launch level is around 260 Pa". Please provide an average altitude for this pressure level.

-The different shades of blue and red in Figs 3, 4 and 5 are not always easy to distinguish (maybe it is a problem with my printed copy). You could consider adding black labeled contours to improve legibility.

-Page 7, line 25. "A more closer examination". Please remove "more".

-Figure 4. These temperatures are daily and zonally averaged, or shown instead at a given local time? Please mention it in the figure caption.

-The comparison with the observed seasonal variability would be eased if Figs. 4 and 5 used the solar longitude Ls as a measure of time, instead of the Sol number. At least, please consider adding an additional horizontal axis displaying Ls.

-Page 8, line 29. "During southern winter solstices" → solstice

-The text states (page 5, line 8) that " instantaneous values of the parameterized (unresolved by the model) temperature disturbances $T'$ are impossible to determine" so that an average value $|T'|$ is used instead. However, in all later mentions to these temperature perturbations, $T'$ is used, and not $|T'|$ (e.g. eq. (2), page 7 line 27, page 8 line 18, labels in Figures 3 and 5,...). Please be consistent with the nomenclature across the paper.

-Page 11, lines 5-6: "without subgrid-scale effects effect included". Please remove "effect"

---

## Referee Comment (RC2) · Anonymous Referee #2 · 7 Sep 2018

Review of "Influence of gravity waves on the climatology of high altitude Martian carbon dioxide ice clouds" by Yiğit et al.

General comments

This article presents a modeling study of the influence of gravity waves on the formation of CO2 ice clouds. The signature of gravity waves has been observed in the Martian atmosphere and modeling has shown that they can have a large impact on the thermal structure of the upper atmosphere. This work links the cooling due to gravity wave breaking/saturation with the probability of formation of CO2 ice clouds.

As most climate models of the Mars atmosphere have a horizontal resolution larger than the scale of gravity waves, a parameterization is used to represent these sub-

grid scale effects on the model winds and temperature. The MPI-MGCM is used for the study, which includes a whole atmosphere gravity wave parameterization and a scheme for CO2 condensation/sublimation.

This is a continuation of the work done in Yiğit et al. (2015), extending the analysis from of Ls 0-20° to the rest of the year.

The main aspects of the model and methodology are presented clearly, followed by model results for the first half of the year. The analysis is continued for the full year focusing on results at the 80, 100 and 120 km levels.

The first reviewer has already brought up an important point about the discussion and comparison with observations. It should be more clearly stated that there are very few positively identified observations of CO2 clouds in the second half of the year (and also above 90 km). That does not make this modeling study irrelevant; it is still useful to the community and is a small step towards a better understanding of gravity waves and the formation of CO2 ice clouds in the Martian atmosphere.

Specific comments

I agree with the first reviewer that perhaps the Sefton-Nash et al. (2013) paper is not useful to discuss the results for the second half of the year as that study was not able to distinguish the aerosol type and in fact, found that nearby temperatures tended to be warmer than the CO2 ice threshold. As summarized in Gonzalez-Galindo et al. (2011), some mid-latitude clouds were seen between Ls 200-300° by THEMIS-VIS, but again, the composition could not be determined. Also, maybe the SPICAM stellar occultation measurements by Montmessin et al. (2006) should be in the list of observations for completeness.

The lack of full diurnal coverage of observations makes this kind of comparison difficult. It might be useful to look at daytime and nighttime averages of the shown model quantities (as was done in Yiğit et al., 2015) to better understand the high probabilities

in the second half of the year and at the higher altitudes.

In terms of the discussion, the possible reasons for discrepancies were well presented, for example, the uncertainty in sources and the degree of supersaturation. Two other possible uncertainties perhaps could also be mentioned, one is the radiative impact of water ice clouds in the first half of the year and the other is the vertical distribution of dust in the second half of the year. Both are lower atmosphere phenomenon but do affect the strength of the global circulation patterns. It might be useful to discuss how sensitive the parameterization is to these effects. This may also help to explain discrepancies seen in the comparison of temperatures at 80 km to MCS (figure 4a with Sefton-Nash 2013 figure 10).

In section 2.2 (page 4), it is mentioned that 'This formulation requires also a prescription of the characteristic horizontal scale $\lambda_h$ of GWs for calculating $\tau_i$', it might be useful to state what is used for this study. Is this value a source of uncertainty as well?

Below are my minor technical comments:

Page 1 line 13: May I suggest: "Thus, Mars has seasons similar to those one is familiar with on Earth."

Page 2 line1: 'on average,' and yes, warmer than what?

Page 2 line 10: suggestion: 'with the exception of harmonics with zero horizontal phase velocities with respect to the surface generated by the flow over topography'

Page 5 line 18: "P must be treated as a certain metric introduced"

Figures: Agree with reviewer 1, figures 4 and 5 x-axis label in Ls would be more useful than day number.

Figures 3c,d and 5a,b,c some contour lines to help distinguish?

Figure 4a very difficult (almost impossible) to compare with Sefton-Nash et al., 2013 figure 10. A change in color scale to match would be useful.

---

## Author Comment (AC1) · 9 Oct 2018

**Response to the comments of the Reviewer 1:**

**General comments**

1. The paper is well written and easy to read. The topic is of interest for the Martian community (and also for people studying high altitude clouds on other planets, including the Earth). The introduction adequately summarizes previous works on the topic and sets up the questions raised by them. The figures are appropriate, although some minor improvements could be made to facilitate the comparison with observations (see below). The length of the manuscript is also appropriate. However, I find the discussion about the comparison with the observations to be flawed due to the incorrect assumptions the authors make about the observations in *Sefton-Nash et al.* [2013] (see details below). This is an essential aspect of the manuscript that absolutely needs to be corrected before publication.

   We thank the reviewer for insightful and useful comments. In the manuscript and the response letter concerns of the Reviewer have been addressed.

**Specific comments**

1. The comparison between the observed cloud climatology and the model predictions should be based only on clouds spectroscopically unambiguously determined to be composed of CO2. The differences between the observed and predicted CO2 cloud climatology should be acknowledged, and the implications for the gravity wave activity in the model discussed.
   This indeed was the point, which we did not fully realize at the time of writing. We thank both Reviewers for bringing to us this important point. The text and discussion have been modified accordingly.

2. Another interesting aspect that deserves further discussion is the altitude variation of the cloud formation probability, predicted to be significantly larger at 120 km than at 80 km. Although the altitude of CO2 mesospheric clouds is not easy to determine for most of the datasets, the current observational knowledge is that, at least during daytime, they are placed at altitudes of about 70-80 km [*Scholten et al.*, 2010; *Määttänen et al.*, 2010]. During nighttime SPICAM has detected mesospheric clouds with altitudes around 100 km. No clouds have been detected, to my knowledge, around 120 km or higher, where the model predicts the higher cloud formation probability. I would like to see a discussion about this discrepancy in the manuscript.
   We already discussed the importance of explicitly considering microphysics of cloud formation. We hypothesize two possibilities: either clouds are not formed in the upper atmosphere due to the lack of nuclei or for other microphysical reasons, or they are formed but not detected being too thin and/or having too small particle sizes. This, of course, if the simulated GW-induced cold pockets are trusted, as we believe they are now. The text have been extended to reflect this point.

3. The GCM used in the study is very shortly described, apart from the gravity wave parameterization. While this is mostly OK given that the model has already been described in previous papers, I think the implementation of the physical processes affecting the temperatures in the mesosphere/lower

[Figure]

Figure 1: Comparison of 1-D volume mixing profiles, grey area indicates 1 standard deviation. The IUVS/MAVEN retrieval is given by the blue line, green and yellow ones are for the *Nair et al.* [1994] (N94) and Mars Climate Database (MCD) profiles based on photochemical modeling. From [*Mockel et al.*, 2017].

thermosphere needs to be described to some extent. For example, what atomic oxygen distribution are you using among the different possibilities discussed in *Medvedev et al.* [2015]?

We have somewhat extended the model description.

In particular, we used the so-called *Nair et al.* [1994] scenario for the vertical distribution of atomic oxygen. The rationale for that is the following. Recently, retrievals of [O] from airglow emission measurements by Imaging Ultraviolet Spectrograph onboard the MAVEN orbiter (IUVS/MAVEN) became available. They have insufficient spatial and temporal resolution, however they allowed for constructing an oxygen scenario by averaging into a single profile [*Mockel et al.*, 2017]. The comparison between thus obtained "IUVS" scenario and the two discussed in the paper of *Medvedev et al.* [2015] is shown in Figure 1. It is seen that the IUVS oxygen profile lies approximately between the other two available scenarios. However, the IUVS/MAVEN observations were performed during daytime only. It is well known that the production of [O] maximizes on the Sun-lit side as well as its concentration. Therefore, it is plausible to assume that the day time-based IUVS scenario overestimated concentrations, and the daily averaged value would be closer to the *Nair et al.* [1994] scenario.

4. Page 4, lines 8-9. "In the middle and upper atmosphere of Mars, wave damping occurs due primarily to nonlinear wave-wave interactions (breaking and/or saturation) and molecular diffusion and thermal conduction, which are accounted for through the transmissivity". Eckermann et al., Icarus 211, pp. 429-442 (2011) showed that radiative damping can be a dominant process in the middle atmosphere of Mars. Do you consider it in your model?

[Figure]

Figure 2: Vertical profiles calculated with the column GW model for a spectrum of GWs with horizontal phase velocities between 0 and 60 m s$^{-1}$: (a) Rms fluctuations of the GW-induced horizontal wind without accounting for $CO_2$ radiative cooling; (b) Difference (negative) in the amplitude (rms) produced by the included radiative damping.

The influence of the radiative damping by $CO_2$ has been studied and results described in our earlier paper [*Medvedev et al.*, 2011, Section 7]. There, the radiative damping rates $\tau_d^{-1}$ calculated by Eckermann have been implemented into our parameterization of GWs, which is used in the current study. The results did not confirm quantitatively the hypothesis of Eckermann. They revealed that $CO_2$ does produce some GW damping in the lower and middle atmosphere, but its effect on wave amplitudes and the produced drag is small. Figure 11(j,k,l) of the paper by *Medvedev et al.* [2011] demonstrates that the radiative damping has only minor net impact on the fields simulated with the MAOAM MGCM. Since wave amplitudes are important within the context of the current study, we attach Figure 2 showing a typical profile of GW activity (rms wind fluctuations for a spectrum of waves) calculated with a column model. These results were closely discussed with Dr. Steve Eckermann. Note that Steve does not pursue his hypothesis since 2011.
The answer to your question is "No, we do not include radiative damping by $CO_2$ in our simulations."

5. Figure 1. The gravity wave cooling is generally below 60 K/day except for the strong peak at 140 km reaching 120 K/day. Could you provide an the explanation for this strong peak? It apparently affects only one or two model layers, could you confirm this? Can it be due to any boundary effect? Note also that Fig. 1b) horizontal axis is labeled as K/day, but in the caption you state it is in units of K/sol, which is slightly different. Please correct.
This zigzag appeared due to differentiating at he top two model layers and was caused by the

plotting software - the "non-existing" data were treated as having particular numbers. The plot is now corrected along with the label "K/sol". All values throughout the paper are assumed to be in sols. We have fully revised Figure 1 and fixed plotting bugs.

6. Page 8, lines 4-5: "The model generally reproduces the observed temperature well, except that it overestimates it in the southern hemisphere winter by up to 20 K". Other data-model discrepancies are evident by comparison of Fig. 4a with Figure 10 in Sefton-Nash et al., (2013). In particular, the temperature at 80 km in the polar regions can be higher than 180 K in MCS observations, while apparently (but this is maybe an artifact of the chosen color scale) do not go much higher than 150 K in the model. Could you please clarify it?

This is indeed an artifact of plotting in Grads software that we have originally used. We have redone Figure 4 and added higher temperature shading levels in the figure. Now, it can be seen that the temperatures in the polar regions can occasionally exceed 175 K, showing a reasonable agreement with MCS.

7. -Page 8, lines 10-11: "It is seen that the coldest temperatures of down to 90-100 K are found around the summer high-latitudes at solstices and during equinoxes". I do not see those low temperatures during equinoxes, when apparently temperatures do not go below 120 K, as can be seen also in Fig. 2a. Please clarify/correct.

Yes, this indeed is obvious from Figure 4c. The words "and during equinoxes" have been removed.

8. -Page 9, lines 14-16. "Although the vast majority of studies report on cloud observations in the Martian mesosphere below 80 km, there are some studies that extend their analysis to higher altitudes presenting detections of CO2 clouds at around the mesopause ( 100 km) and above (e.g. Sefton-Nash et al., 2013)". Sefton-Nash et al. (2013) only detected clouds up to 90 km (e.g. their figure 9).

We have now removed this statement.

**Technical Comments:**

1. Page 2, line 1: "Because the Martian mesosphere is, in average, warmer ..." Warmer than the terrestrial one, or warmer than the CO2 frost point? Please specify.

Here we refer to the mean temperature in comparison with the condensation threshold. However, there are significant variations around the mean. So, one way of explaining the occurrence of thermodynamically favorable conditions for cloud formation is that there are variations induced by tides and gravity waves around the mean temperature. This is now clear in the text.

2. Page 3, line 5 "variations variations". Please remove one.

Done

3. Page 3, line 30: It was developed in detail in the work of Yigit et al. (2008), the general principles of .... I think either removing "in detail" or changing to "described in detail" would be more correct. Also please add "and" after the comma.

Done

4. Page 4, lines 29-30. This launch level is around 260 Pa. Please provide an average altitude for this pressure level.

    It is $\sim$ 8 km. Added in the text.

5. The different shades of blue and red in Figs 3, 4 and 5 are not always easy to distinguish (maybe it is a problem with my printed copy). You could consider adding black labeled contours to improve legibility.

    We have redone these figures and tried to improved the quality. We have now confirmed that in the printed version, the contours are clearly visible.

6. Page 7, line 25. A more closer examination. Please remove more.

    Done

7. Figure 4. These temperatures are daily and zonally averaged, or shown instead at a given local time? Please mention it in the figure caption.

    Added in the caption.

8. The comparison with the observed seasonal variability would be eased if Figs. 4 and 5 used the solar longitude Ls as a measure of time, instead of the Sol number. At least, please consider adding an additional horizontal axis displaying Ls.

    We have replaced the axis to the solar longitude, which is more useful, as the referee pointed out.

9. Page 8, line 29. "During southern winter solstices" $\rightarrow$ solstice

    Done.

10. The text states (page 5, line 8) that "instantaneous values of the parameterized (unresolved by the model) temperature disturbances $T'$ are impossible to determine" so that an average value $|T'|$ is used instead. However, in all later mentions to these temperature perturbations, $T'$ is used, and not $|T'|$ (e.g. eq. (2), page 7 line 27, page 8 line 18, labels in Figures 3 and 5,...). Please be consistent with the nomenclature across the paper.

    Consistency is now provided in the entire text.

11. Page 11, lines 5-6: "without subgrid-scale effects effect included". Please remove "effect"

    Removed .

**References**

Medvedev, A. S., E. Yiğit, P. Hartogh, and E. Becker (2011), Influence of gravity waves on the Martian atmosphere: general circulation modeling. *J. Geophys. Res. Planets*, *116*, E10004, doi:10.1029/2011JE003848.

Medvedev, A. S., F. González-Galindo, E. Yiğit, A. G. Feofilov, F. Forget, and P. Hartogh (2015), Cooling of the Martian thermosphere by CO2 radiation and gravity waves: An intercomparison study with two general circulation models, *J. Geophys. Res. Planets*, *120*, 913–927, doi:10.1002/2015JE004802.

Mockel, C., A. S. Medvedev, P. Hartogh, E. Yiğit and the MAVEN team (2017), Martian thermosphere in MAVEN/IUVS data and MPI-MGCM, *Sixth international workshop on the Mars atmosphere: Modelling and observations*, Granada, January 17–20, 2017, http://www-mars.lmd.jussieu.fr/granada2017/abstracts/mockel_granada2017.pdf

Nair, H., M. Allen, A. D. Anbar, Y. L. Yung, and R. T. Clancy (1994), A photochemical model of the martian atmosphere, *Icarus*, *111*, 124–150.

Määttänen, A., F. Montmessin, B. Gondet, F. Scholten, H. Hoffmann, F. González-Galindo, A. Spiga, F. Forget, E. Hauber, G. Neukum, J.-P. Bibring, and J.-L. Bertaux (2010), Mapping the mesospheric CO2 clouds

on mars: MEx/OMEGA and MEx/HRSC observations and challenges for atmospheric models, *Icarus*, *209*(2), 452–469, doi:http://dx.DOI.org/10.1016/j.icarus.2010.05.017.

**Scholten, F., H. Hoffmann, A. Määttänen, F. Montmessin, B. Gondet, and E. Hauber** (2010), Concatenation of HRSC colour and OMEGA data for the determination and 3d-parameterization of high-altitude co2 clouds in the martian atmosphere, *Planet. Space Sci.*, *58*, doi: 10.1016/j.pss.2010.04.015.

**Sefton-Nash, E., N. A. Teanby, L. Montabone, P. G. J. Irwin, J. Hurley, and S. B. Calcutt** (2013), Climatology and first-order composition estimates of mesospheric clouds from mars climate sounder limb spectra, *Icarus*, *222*, 342–356, doi:10.1016/j.icarus.2012.11.012.

---

## Author Comment (AC2) · 9 Oct 2018

The comment was uploaded in the form of a supplement:
https://www.ann-geophys-discuss.net/angeo-2018-61/angeo-2018-61-AC2-supplement.pdf

---

## Author Comment (AC3) · 9 Oct 2018

**Response to the comments of the Reviewer 2:**

**Specific Comments:**

1. I agree with the first reviewer that perhaps the *Sefton-Nash et al.* [2013] paper is not useful to discuss the results for the second half of the year as that study was not able to distinguish the aerosol type and in fact, found that nearby temperatures tended to be warmer than the CO2 ice threshold. As summarized in *González-Galindo et al.* [2011], some mid-latitude clouds were seen between $L_s$ $200 - 300°$ by THEMIS-VIS, but again, the composition could not be determined.

   We modified the text to explicitly admit that the composition of the majority of clouds observed in the second half of the year is not established, and that they are likely not $CO_2$. However, we report on what our simulations have produced. Namely, that gravity wave-induced temperature fluctuations can create conditions favorable for ice condensation throughout the second half of the year as well.

2. The lack of full diurnal coverage of observations makes this kind of comparison difficult. It might be useful to look at daytime and nighttime averages of the shown model quantities (as was done in *Yiğit et al.* [2015] to better understand the high probabilities in the second half of the year and at the higher altitudes.

   The main reason is the increased gravity wave activity in middle-to-high latitudes of the winter hemispheres at both aphelion and perihelion seasons (see the first row in Figure 5). It is also seen that GW-induced temperature fluctuations are larger during the second half of the year. However, given the lack of observational constraints for $CO_2$ clouds in the second half of the year, we felt it would be rather senseless and speculative to go into detailed discussions of the diurnal structure of the simulated cloud probabilities.

3. In terms of the discussion, the possible reasons for discrepancies were well presented, for example, the uncertainty in sources and the degree of supersaturation. Two other possible uncertainties perhaps could also be mentioned, one is the radiative impact of water ice clouds in the first half of the year and the other is the vertical distribution of dust in the second half of the year. Both are lower atmosphere phenomenon but do affect the strength of the global circulation patterns. It might be useful to discuss how sensitive the parameterization is to these effects. This may also help to explain discrepancies seen in the comparison of temperatures at 80 km to MCS (figure 4a with Sefton-Nash 2013 figure 10).

   The uncertainties mentioned in the manuscript relate primarily to the physics phenomena, which are the main focus here: gravity waves and $CO_2$ condensation. However, we do agree that other factors affect the simulated circulation and thus contribute to the discrepancies as well. We added the corresponding wording into the text. Exploring the sensitivity of the simulated circulation to the missing in the model mechanisms is beyond the scope of this paper, and requires a separate dedicated study.

4. In section 2.2 (page 4), it is mentioned that "This formulation requires also a prescription of the characteristic horizontal scale $\lambda_h$ of GWs for calculating $\tau_i$", it might be useful to state what is used for this study. Is this value a source of uncertainty as well?

   We added the information on the characteristic wavelength $\lambda_h = 300$ km utilized in the gravity wave

parameterization. This value is also a source of uncertainties, like with any parameterization. However, in this particular case, the uncertainty is small, as the appropriate characteristic wavelegths are limited by the 100-500 km range. The reason for that and consequences were discussed in detail in many of our referenced papers on the gravity wave parameterization.

**Technical Comments:**

1. Page 1 line 13: May I suggest: "Thus, Mars has seasons similar to those one is familiar with on Earth."
   Done.

2. Page 2 line 1: "on average," and yes, warmer than what? "...warmer than the condensation threshold". Now added.

3. Page 2 line 10: suggestion: "with the exception of harmonics with zero horizontal phase velocities with respect to the surface generated by the flow over topography"
   Suggestion implemented

4. Page 5 line 18: "P must be treated as a certain metric introduced".
   Done

5. Figures: Agree with reviewer 1, figures 4 and 5 x-axis label in Ls would be more useful than day number.
   Done

6. Figures 3c,d and 5a,b,c some contour lines to help distinguish?
   Figure redone

7. Figure 4a very difficult (almost impossible) to compare with Sefton-Nash et al., 2013 figure 10. A change in color scale to match would be useful.
   Now it can be better compared

**References**

**González-Galindo, F., A. Määtänen, F. Forget, and A. Spiga** (2011), The martian mesosphere as revealed by $CO_2$ cloud observations and general circulation modeling, *Icarus*, *216*, 10–22, doi:10.1016/j.icarus.2011.08.006.

**Sefton-Nash, E., N. A. Teanby, L. Montabone, P. G. J. Irwin, J. Hurley, and S. B. Calcutt** (2013), Climatology and first-order composition estimates of mesospheric clouds from mars climate sounder limb spectra, *Icarus*, *222*, 342–356, doi:10.1016/j.icarus.2012.11.012.

**Yiğit, E., A. S. Medvedev, and P. Hartogh** (2015), Gravity waves and high-altitude $CO_2$ ice cloud formation in the martian atmosphere, *Geophys. Res. Lett.*, *42*, doi: 10.1002/2015GL064275.

---

## Author Comment (AC4)

[revised manuscript text omitted]

- 20 *kt/dt/studbes/th/dt/studbes/th/dt/studbes/pt/kk/dt/dt/studbes/pt/kk/dt/dt/studbes/pt/kk/dt/dt/studbes/pt/kk/dt/dt/studbes/pt/kk/dt/dt/studbes/pt/kk/dt/dt/studbes/pt/kk/dt/dt/studbes/pt/kk/dt/studbes/pt/kk/dt/studbes/pt/kk/dt/studbes/pt/kk/dt/studbes/pt/kk/dt/studbes/pt/kk/dt/studbes/pt/kk/dt/studbes/pt/kk/dt/studbes/pt/kk/dt/studbes/pt/kk/dt/studbes/pt/kk/dt/studbes/pt/kk/dt/studbes/pt/kk/dt/studbes/pt/kk/dt/studbes/pt/kk/dt/studbes/pt/kk/dt/studbes/pt/kk/dt/studbes/pt/kk/dt/studbes/pt/kk/dt/studbes/pt/kk/dt/studbes/pt/kk/dt/studbes/pt/kk/dt/studbes/pt/kk/dt/studbes/pt/kk/dt/studbes/pt/kk/dt/studbes/pt/kk/dt/studbes/pt/kk/dt/studbes/pt/kk/dt/studbes/pt/kk/dt/studbes/pt/kk/dt/studbes/pt/kk/dt/studbes/pt/kk/dt/studbes/pt/kk/dt/studbes/pt/kk/dt/studbes/pt/kk/dt/studbes/pt/kk/dt/studbes/pt/kk/dt/studbes/pt/kk/dt/studbes/pt/kk/dt/studbes/pt/kk/dt/studbes/pt/kk/dt/studbes/pt/kk/dt/studbes/pt/kk/dt/studbes/pt/kk/dt/studbes/pt/kk/dt/studbes/pt/kk/dt/studbes/pt/kk/dt/studbes/pt/kk/dt/studbes/pt/kk/dt/studbes/pt/kk/dt/studbes/pt/kk/dt/studbes/pt/kk/dt/studbes/pt/kk/dt/studbes/pt/kk/dt/studbes/pt/kk/dt/studbes/pt/kk/dt/studbes/pt/kk/dt/studbes/pt/kk/dt/studbes/pt/kk/dt/studbes/pt/kk/dt/studbes/pt/kk/dt/studbes/pt/kk/dt/studbes/pt/kk/dt/studbes/pt/kk/dt/studbes/pt/kk/dt/studbes/pt/kk/dt/studbes/pt/kk/dt/studbes/pt/kk/dt/studbes/pt/kk/dt/studbes/pt/kk/dt/studbes/pt/kk/dt/studbes/pt/kk/dt/studbes/pt/kk/dt/studbes/pt/kk/dt/studbes/pt/kk/dt/studbes/pt/kk/dt/studbes/pt/kk/dt/studbes/pt/kk/dt/studbes/pt/kk/dt/studbes/pt/kk/dt/studbes/pt/kk/dt/studbes/pt/kk/dt/studbes/pt/kk/dt/studbes/pt/kk/dt/studbes/pt/kk/dt/studbes/pt/kk/dt/studbes/pt/kk/dt/studbes/pt/kk/dt/studbes/pt/kk/dt/studbes/pt/kk/dt/studbes/pt/kk/dt/studbes/pt/kk/dt/studbes/pt/kk/dt/studbes/pt/kk/dt/studbes/pt/kk/dt/studbes/pt/kk/dt/studbes/pt/kk/dt/studbes/pt/kk/dt/studbes/pt/kk/dt/studbes/pt/kk/dt/studbes/pt/kk/dt/studbes/pt/kk/dt/studbes/pt/kk/dt/studbes/pt/kk/dt/studbes/pt/kk/dt/studbes/pt/studbes/pt/studbes/pt/studbes/pt/studbes/pt/studbes/pt/studbes/pt/stu*

[revised manuscript text omitted]

---

## Referee Report (RR1)

2nd Review of "Influence of gravity waves on the climatology of high altitude Martian carbon dioxide ice clouds" by Yiğit et al.

The manuscript has been adapted sufficiently to address the comments and questions and the figures are much clearer.

Only a very minor comment regarding the added line on page 4, line 17: 'multiply' doesn't really fit, it is enough to say 'For the reasons described in our papers' and I think the reference is Medvedev et al, 2011a. (The 2011b paper uses 200 km as stated in Section 4, 1st paragraph.)

---

## Referee Report (RR2)

**Review of the manuscript "Influence of gravity waves on the climatology of high-altitude Martian carbon dioxide ice clouds" by Yiğit et al., submitted to Annales Geophysicae**

I am generally satisfied with the modifications implemented by the authors in the revised version of the manuscript. I particularly appreciate the efforts to improve the quality and readability of the figures, which I think are significantly better now. However, I still have problems with the statements made about the comparison with observations, which was the main point in my previous comment.

The authors have introduced an statement acknowledging that the observations in Sefton-Nash et al. (2013) do not discriminate between $CO_2$ and $H_2O$ clouds (page 9, lines 26-31 of the revised manuscript), which is good. But it is important to note that Sefton-Nash et al. (2013) use MCS temperature measurements to find that clouds detected during the second half of the year are not $CO_2$ clouds, but very likely $H_2O$ clouds of even dust. However, the authors insist in comparing the observations in Sefton-Nash et al. (2013) during the second half of the year ($H_2O$ clouds) with their derived probability of $CO_2$ cloud formation. I think that sentences such as "The model reproduces more favorable conditions for $CO_2$ condensation in the midlatitudes regions during wintertime. It agrees with observations in that mesospheric clouds occur more frequently during perihelion (Figure 5g)" (page 10, lines 7-9 of the revised manuscript) are misleading, as they are not comparing apples to apples. $CO_2$ cloud formation probability should not be compared to $H_2O$ clouds observations, as the conditions for forming $CO_2$ and $H_2O$ clouds are very different.

In my opinion, the two first paragraphs in page 10 still need to be rewritten, focusing the comparison only on the observed distribution of unambiguously detected $CO_2$ clouds. In particular, I think it should be explicitly acknowledged that the prediction of elevated $CO_2$ cloud formation during the second half of the year does not agree with observations of $CO_2$ clouds. Note that, in my opinion, this difference does not decrease in any point the merits of the model. Often the most interesting discoveries come from differences between observations and models. It is also possible that there biases in the observations, and the predictions of the model may motivate future observational searches for clouds in the regions/seasons pointed by the model. The authors already discuss possibilities for the model/observation differences, and possible limitations of the model, so no additional discussion would be needed (although I think the use of a constant O density profile for all conditions, seasons, latitudes and Local Times could be considered another limitation affecting the seasonal variability of the temperatures).

---

## Author Response (AR2)

**Response to the comments of the Reviewer 1: (Revision 2)**

**General comments**

1. I am generally satisfied with the modifications implemented by the authors in the revised version of the manuscript. I particularly appreciate the efforts to improve the quality and readability of the figures, which I think are significantly better now. However, I still have problems with the statements made about the comparison with observations, which was the main point in my previous comment.

   The authors have introduced an statement acknowledging that the observations in *Sefton-Nash et al.* [2013] do not discriminate between CO2 and H2O clouds (page 9, lines 26-31 of the revised manuscript), which is good. But it is important to note that Sefton-Nash et al. (2013) use MCS temperature measurements to find that clouds detected during the second half of the year are not CO2 clouds, but very likely H2O clouds of even dust. However, the authors insist in comparing the observations in Sefton-Nash et al. (2013) during the second half of the year (H2O clouds) with their derived probability of CO2 cloud formation. I think that sentences such as "The model reproduces more favorable conditions for CO2 condensation in the midlatitudes regions during wintertime. It agrees with observations in that mesospheric clouds occur more frequently during perihelion (Figure 5g)" (page 10, lines 7-9 of the revised manuscript) are misleading, as they are not comparing apples to apples. CO2 cloud formation probability should not be compared to H2O clouds observations, as the conditions for forming CO2 and H2O clouds are very different.

   In my opinion, the two first paragraphs in page 10 still need to be rewritten, focusing the comparison only on the observed distribution of unambiguously detected CO2 clouds. In particular, I think it should be explicitly acknowledged that the prediction of elevated CO2 cloud formation during the second half of the year does not agree with observations of CO2 clouds. Note that, in my opinion, this difference does not decrease in any point the merits of the model. Often the most interesting discoveries come from differences between observations and models. It is also possible that there biases in the observations, and the predictions of the model may motivate future observational searches for clouds in the regions/seasons pointed by the model. The authors already discuss possibilities for the model/observation differences, and possible limitations of the model, so no additional discussion would be needed (although I think the use of a constant O density profile for all conditions, seasons, latitudes and Local Times could be considered another limitation affecting the seasonal variability of the temperatures).

   We agree with the reviewer that direct comparison of the simulations and observations published by *Sefton-Nash et al.* [2013] is not completely possible, because the nature of clouds detected in the second half of the year is not known. Our point indeed required elaboration along the following lines. Water ice clouds usually do not extend higher than 40 km except during perihelion, when they rise up to 60-65 km. Therefore, all clouds observed above 70 km are likely not water ice clouds. The question regarding the nature of these clouds is still open. The arguments that they were not $CO_2$ are based on the fact that the mean temperature was too warm to sustain condensation. Our simulations provide arguments that, at least part of the elevated clouds, could be $CO_2$ clouds, if localized gravity wave-induced disturbances of

temperature are accounted for. Overall, we would like to retain the structure of our discussion on page 10 with further clarifications.

The vast majority of the explicit observations of $CO_2$ clouds provide a very limited seasonal coverage as the observations were relatively unevenly sampled. Therefore, considerations of daytime clouds only, which a lot of observations are limited to, includes already an observational bias in the (seasonal) distribution of these clouds.

We appreciate the comment of the reviewer about the constant density of atomic oxygen. We have now acknowledged this limitation in discussion on page 11.

   Thanks for this corrections. It is now implemented.

[revised manuscript text omitted]